# Engineering of MSC-Derived Exosomes: A Promising Cell-Free Therapy for Osteoarthritis

**DOI:** 10.3390/membranes12080739

**Published:** 2022-07-28

**Authors:** Jin Cheng, Yixin Sun, Yong Ma, Yingfang Ao, Xiaoqing Hu, Qingyang Meng

**Affiliations:** 1Department of Sports Medicine, Peking University Third Hospital, Institute of Sports Medicine of Peking University, Beijing Key Laboratory of Sports Injuries, Beijing 100191, China; chengjin@bjmu.edu.cn (J.C.); huidong01@sina.com (Y.M.); aoyingfang@163.com (Y.A.); 2Peking Unversity First Hospital, Peking University Health Science Center, Beijing 100034, China; 1810301146@pku.edu.cn

**Keywords:** exosomes, mesenchymal stem cell, engineering, cartilage regeneration, osteoarthritis

## Abstract

Osteoarthritis (OA) is characterized by progressive cartilage degeneration with increasing prevalence and unsatisfactory treatment efficacy. Exosomes derived from mesenchymal stem cells play an important role in alleviating OA by promoting cartilage regeneration, inhibiting synovial inflammation and mediating subchondral bone remodeling without the risk of immune rejection and tumorigenesis. However, low yield, weak activity, inefficient targeting ability and unpredictable side effects of natural exosomes have limited their clinical application. At present, various approaches have been applied in exosome engineering to regulate their production and function, such as pretreatment of parental cells, drug loading, genetic engineering and surface modification. Biomaterials have also been proved to facilitate efficient delivery of exosomes and enhance treatment effectiveness. Here, we summarize the current understanding of the biogenesis, isolation and characterization of natural exosomes, and focus on the large-scale production and preparation of engineered exosomes, as well as their therapeutic potential in OA, thus providing novel insights into exploring advanced MSC-derived exosome-based cell-free therapy for the treatment of OA.

## 1. Introduction

Osteoarthritis (OA), characterized by progressively escalating cartilage deterioration, is a degenerative joint disease with a growing prevalence and disease burden worldwide [1]. OA can cause pain, limited movement, swelling and joint instability, which all ultimately lead to physical disability and reduced quality of life [2]. Currently, OA is mainly managed through non-pharmacological, pharmacological and surgical therapies [3]. However, many challenges remain to be addressed in OA treatment, including patient compliance, adverse effects of medications and surgical complications [4,5,6]. On the one hand, the pathogenesis of OA is fairly complex, involving progressive destruction of articular cartilage with changes in composition and physical properties, as well as subchondral bone remodeling and synovial inflammation [1]. On the other hand, instead of cartilage repair and regeneration, managing disease-associated symptoms remains the goal of current OA treatment in most cases [5]. Despite temporary relief of the symptoms, the overall effect of current treatments is unsatisfactory because the progression of OA is not reversed.

Mesenchymal stem cells (MSCs) are pluripotent stem cells with differentiation potential and immunomodulatory properties, and have emerged as a promising cell-based therapy for OA [7,8,9]. The effects of MSCs on OA is mainly reflected in the enhancement of chondrogenesis and proliferation, the reduction of apoptosis and the maintenance of autophagy of chondrocytes [10]. Mechanistically, mitogen-activated protein kinase (MAPK), Wnt and Notch signaling pathways are involved in the MSC-induced chondrogenesis [11]. MSCs also regulate inflammatory cytokines including interferon-gamma (IFN-γ) and prostaglandin E2 to exert anti-inflammatory effects, and release growth factors to promote tissue repair, such as vascular endothelial growth factor (VEGF) and transforming growth factor-beta (TGF-β) [11].

Similar to MSCs, the use of their secretome (growth factors, cytokines, chemokines, enzymes, lipids, nucleic acids and transcription factors) and extracellular vesicles (either microvesicles or exosomes) has pleiotropic effects on OA arthrosis, such as immunomodulatory, regenerative, anti-catabolic and chondro-protective properties [12]. Furthermore, accumulating evidence have demonstrated that the therapeutic efficacy of MSCs in OA relies mostly on the paracrine function of MSCs, especially the secretion of exosomes [13,14,15]. Given that the safety concerns regarding the toxicity, biodistribution and potential tumorigenicity of MSCs are still unresolved, exosomes have attracted more and more attention as a promising alternative to MSCs [16].

The concept of “exosomes” was first proposed in 1981 by Trams [17]. Exosomes are extracellular vesicles with a diameter of 30–150 nm that can carry various proteins, lipids and nucleic acid for intercellular communication [18]. It has been shown that exosomes can exhibit diverse regulatory functions in myocardial injury [19], bacterial infection [20], skin diseases [21], diabetic retinopathy [22], hepatocellular carcinoma [23] and neuroblastoma [24]. Since exosomes are endogenously produced, they can avoid immune responses. In addition, they have no tumorigenicity and are easier to store than MSCs [18]. Mesenchymal stem cell-derived exosomes (MSC-Exos) exert therapeutic effects on restoring the structural and functional integrity of articular cartilage and alleviating OA, thus being considered as a promising cell-free tissue engineering therapy [15,18,25,26,27,28,29,30,31]. Although exosomes have their own benefits in terms of tissue regeneration, they still have disadvantages such as low yield, weak function and low targeting, which fail to meet quantity and quality needs for disease treatment. A series of engineering approaches have been developed to overcome the limitations of natural exosomes and advance exosome therapy into clinical practice. The engineering of exosomes can be approached from the perspectives of both parental cells and exosomes themselves, from drug loading to surface modification. However, engineering exosomes still faces challenges including limited drug loading efficiency and insufficient clinical grade production [32]. As a result, there is an urgent need for a set of scalable, high-yield, low-cost producing and processing procedures that can provide exosomes with consistent quality and efficacy.

Here, we aim to review the current knowledge of the biogenesis, isolation and characterization of natural exosomes, and further focus on the large-scale production and processing of engineered exosomes. In addition, current approaches and future perspectives to make use of engineered exosomes as a platform in therapeutic strategies in OA and the associated challenges will be discussed.

## 2. Natural Exosomes

As an indispensable mediator of intercellular communication, exosomes are secreted by almost all kinds of cell types such as B and T lymphocytes, dendritic cells, mast cells, MSCs, platelets and tumor cells [33], and carry various signal substances including DNA, RNA and proteins [34]. Secreted exosomes can interact with recipient cells via surface receptor proteins and through endocytosis or membrane fusion, thereby achieving certain effects [35]. For mass production and purification, it is important to understand the mechanism of biogenesis of exosomes and explore the markers for identification.

### 2.1. Biogenesis of Exosomes

Exosome biogenesis consists of three main stages, which are endosomes, multivesicular bodies (MVBs) and exosomes, involving double invagination of the plasma membrane [36]. The early sorting endosomes (ESEs) arise from the first invagination of the plasma membrane and the entry of cell surface proteins and extracellular components such as lipids, proteins, metabolites, ions and small molecules. Under the regulation of the endosomal sorting complex required for transport (ESCRT) and other proteins, ESEs fuses with the endoplasmic reticulum (ER) and/or the trans-Golgi network (TGN) to form late sorting exosomes (LSEs). The second plasma membrane invagination of LSEs regulated by sorting machineries forms MVBs containing intraluminal vesicles (ILVs). The MVBs can either be degraded by fusing with lysosomes or autophagosomes, or release the contained ILVs as exosomes through exocytosis of plasma membrane [37].

Functions performed by released exosomes depend on the content of exosomes, which is determined by the cytoplasmatic composition of the host cell. Moreover, stress, such as hypoxia, can also affect the content of exosomes. More importantly, the exosomal cargo is regulated by sorting mechanisms. As a well-recognized cargo sorting mechanism, the ESCRT complex includes four sub-complexes, which can identify protein cargoes sequentially and classify them into ILVs [38]. The ESCRT complex controls the entry of ubiquitinated proteins into ILVs and deubiquitination occurs before the entry [38,39]. The ESCRT complex performs three functions: first, it recognizes ubiquitylated cargoes and prevents their recycling and retrograde trafficking; secondly, it deforms the membrane, sorting cargoes into endosomal invaginations; finally, it catalyzes the eventual abscission of the invaginations to form ILVs containing classified cargoes [40]. However, not all the sorting of proteins depends on ubiquitination and the ESCRT complex.

Some microRNAs are enriched in exosomes while others are barely present, highlighting the existence of highly selective RNA sorting mechanisms [41]. Recently, Zhang et al. found that heterogeneous nuclear ribonucleoprotein A1 (hnRNPA1) could mediate miR-522 packing into exosomes [42]. The RNA-induced silencing complex (RISC) component Argonaute 2 (Ago2) could also regulate the sort of microRNAs into exosomes, such as let-7a, miR-100 and miR-320a [43]. Furthermore, membrane proteins such as Vps4A played a role in the sorting of microRNAs into exosomes, the inhibition of which resulted in decreased exosomal miR-92a and miR-150 levels [44].

### 2.2. Characteristic of Exosomes

Exosomes have a unique composition of proteins, RNA and lipids that varies by original cells. More than 286 studies have found 41,860 proteins, >7540 RNA and 1116 lipid molecules that are associated with exosomes [45]. Despite the heterogeneity of exosomes, there are still some common proteins shared by all exosomes from different cell types that can be used as biomarkers for identification. Huang et al. found 138 proteins were stably expressed in exosomes from human adipose-derived stem cells (ADSCs) regardless of the isolation method, including CD109, CD166, HSPA4, TRAP1, RAB2A, RAB11B and RAB14 [46]. Accumulating exosome proteomic studies have found that exosomes contain a series of conserved proteins across species. Almost all exosomes contain antigen presentation proteins (e.g., HLA class I histocompatibility antigen), cell adhesion proteins (e.g., Integrins and Claudin-1), cell structure proteins (e.g., Actins, Myosin and Tublins), heat shock proteins (e.g., Hsp90), metabolic enzymes (e.g., GAPDH), MVB biogenesis associated proteins (e.g., Alix and Tsg101), signaling proteins (e.g., kinases), tetraspanins (e.g., Hsp90), transcription and protein synthesis associated proteins, and trafficking and membrane fusion associated proteins (e.g., annexins and Rab) [47]. The most common exosome-associated protein class is the tetraspanin family, especially CD9, CD63, CD81 and CD82 [47].

### 2.3. Isolation

In order to perform downstream analysis or engineer exosomes, the first step is to obtain a highly purified population of exosomes by appropriate isolation methods. Based on the physiochemical and biological properties of the exosome such as density, size and surface components, several classes of strategies are performed in exosome separation, including ultracentrifugation (UC), size-based isolation techniques, charge neutralization-based polymer precipitation, immunoaffinity and microfluidic techniques (Figure 1) [48].

#### 2.3.1. Ultracentrifugation

UC is still the most commonly used method at present [49], mainly based on the features of exosomes such as size and density. Due to easy operation and a low expertise threshold, UC is popular among researchers and widely used [50]. However, UC is suitable for large sample volumes but not for the small samples from clinical practice due to the time-consuming process and high cost of instruments [51]. When it comes to plasma-derived exosomes, co-isolation with lipoproteins is a major issue because of similar size and density [52]. To improve the purity of exosomes, density gradient centrifugation is often combined with UC [53].

#### 2.3.2. Size-Based Isolation Techniques

Ultrafiltration (UF), which depends on size, is faster and more productive than UC without expensive equipment [54]. However, the membrane pores can be easily blocked [55]. Another technique, size exclusion chromatography (SEC), has been increasingly used since 2016 [49], the application of which is easy, low-cost and quick but fails to separate exosomes from lipoproteins such as UC [52,56]. To overcome the limitation of a one-step isolation procedure based on density or size, a two-step technique combining density cushion separation followed by SEC can efficiently separate exosomes from lipoproteins and plasma proteins [56]. SEC can yield exosomes with higher functionality [57], which is also combined with other methods such as UF and UC [58], to increase the yield and purity of exosomes. Compared to UC, SEC combined with UF generated more exosomes [59].

#### 2.3.3. Charge Neutralization-Based Polymer Precipitation

Polymer precipitation is also a convenient method and does not require any expensive or specialized equipment, such as polyethylene glycol (PEG)-based precipitation [60]. High yield is its characteristic, but it is prone to protein contamination. A study comparing five different protocols for isolating exosomes showed the high yield advantage of the precipitation method; however, the choice of the isolation method should be based on the purpose of downstream experiments [61].

#### 2.3.4. Immunoaffinity

The membrane of exosomes contains some specific proteins, such as CD9, CD63 and CD81, so it is feasible to selectively isolate specific exosomes by microplates and immunobeads. Immunoaffinity is easy to use and cost effective, offering unique advantages such as increased efficiency and specificity of exosome capture, integrity and selective origin of isolated vesicles, and thus it is suitable for clinical samples [62].

#### 2.3.5. Microfluidic Techniques

Microfluidics-based technologies have become important and promising, such as viscoelasticity-based microfluidics [63], automated exosome isolation integrating acoustics and microfluidics [64], microfluidic affinity separation chip [65] and a direct-current insulator-based dielectrophoretic (DC-iDEP) approach [66]. Deterministic lateral displacement (DLD) is a simple, rapid, scalable and automatable technology for exosome isolation [67]. For instance, Joshua et al. developed an integrated nanoscale deterministic lateral displacement (nanoDLD) chip containing 1024 parallel arrays capable of isolating serum and urine exosomes with flow rates up to 900 μL per hour [67].

### 2.4. Characterization

Despite the heterogeneity of exosomes in size, content, function and cellular origin [36], there are still some commonalities that can be used to identify exosomes. Generally, methods such as Western blotting, single particle tracking, electron microscopy, flow cytometry, Raman spectroscopy and surface plasmon resonance (SPR) are performed to characterize exosomes [49]. These methods are mainly divided into two categories: (1) external characterization, including morphology and particle size; (2) inclusion characterization, such as membrane proteins. Similarly, the International Society for Extracellular Vesicles (ISEV) proposed a guideline on minimal experimental analysis for studies of extracellular vesicles, including quantification and single vesicle analysis, and characterization by proteins [68].

#### 2.4.1. External Characterization

For quantification of exosomes, nanoparticle tracking analysis (NTA) can determine particle concentration and size distribution but cannot detect exosomes with the size of <50 nm in diameter [69]. Dynamic light scattering (DLS) is able to detect small particles (>5 nm), which is suitable for measuring monodisperse particles but not for complex exosome samples with a large size range [70]. However, NTA and DLS are unable to distinguish exosomes from protein aggregates or nanoparticles because they rely on particle-induced scattered light rather than particle morphology. In single vesicle analysis, electron microscopy is often used to observe exosome morphology, such as transmission electron microscopy (TEM), scanning electron microscopy (SEM) and cryogenic electron microscopy (cryo-EM). SEM shows the surface of exosomes, TEM reveals the internal structure and Cryo-EM directly analyzes the exosome structure in a near-native state. Atomic force microscopy (AFM) is another method for the simultaneous nanomechanical and morphological analysis of exosomes more efficiently [71].

#### 2.4.2. Inclusion Characterization

When it comes to the proteins for characterization, the guideline highlights three categories of protein markers that must be analyzed [68]. The two main types are transmembrane or GPI-anchored proteins and cytosolic proteins with the ability to bind to membranes or to cytosolic sequences of transmembrane proteins, the presence of which demonstrates the lipid-bilayer structure specific of exosomes. Proteins often co-isolated with exosomes can serve as negative markers to evaluate the purity, such as apolipoproteins and albumin for plasma/serum exosomes [56]. With the help of proteins, we can obtain the purity of the extracted exosomes. Western blotting is the most commonly used method for quantifying proteins in or on exosomes; moreover, flow cytometry, fluorescence scanning, SPR and mass spectrometry are optional methods that should be considered as needed [68].

## 3. Techniques for Large-Scale Exosome Production

MSCs have been known for their capacity of differentiation, and the therapeutic efficacy of MSCs relies mostly on the paracrine function of MSCs [72,73]. MSC-Exos have exhibited great therapeutic potential in OA treatment. However, the ability of cells to secrete exosomes cannot meet the high doses required for disease treatment. For better clinical application, it is necessary for large-scale production, while also reducing the cost of treatment. To overcome the limitation of low production efficiency of exosomes, stimulating the cells or disrupting cell membranes can increase the particle yield per cell, as well as developing large-scale cell culture platforms (Figure 2) [74].

### 3.1. Large-Scale Cell Culture Platforms

The most straightforward way for large-scale production of MSC-Exos is MSC expansion, which can be achieved by increasing surface area for cell growth (Figure 2A). Traditional two-dimensional (2D) cell culture requires a large amount of cell culture medium, materials and space, negatively affecting the potency of MSCs [75]. It can be improved to some extent by using simple planar multilayered stack systems, which increase the cell culture scale at the lower cost [76]. Recently, the trend is turning to three-dimensional (3D) culture combined with scalable bioreactors [77]. 3D culture produces large quantities of exosomes more efficiently than 2D culture [78]. Compared to 2D culture, bioreactor-based 3D culture systems can generate higher quality cell products through comprehensive monitoring and precise control of all culture parameters and reduced manual operations [79]. Representative 3D culture bioreactors for the expansion of MSCs mainly include perfusion-based bioreactors and stirred suspension bioreactors [76].

#### 3.1.1. Stirred Suspension Bioreactors

The key part of stirred suspension bioreactors is the appropriate microcarrier, which is suspended in a stirred vessel [76]. Microcarriers are expandable support surfaces for cell growth, enabling high-level expansion, which can be coated with some substances such as PEG that facilitate cell adhesion and proliferation [80]. A hydrogel-based microcarrier was developed from genipin cross-linked alginate-chitosan beads: the MSCs cultured on which exhibited a 26% higher cell attachment rate, twice the proliferation rate and easy isolation without extended incubation or intense agitation [81]. Krutty et al. developed a chemically defined synthetic copolymer coating called PVG (poly(poly(ethylene glycol) methyl ether methacrylate-ran-vinyl dimethyl azlactone-ran-glycidyl methacrylate), P(PEGMEMA-r-VDM-r-GMA)) and validated that MSCs cultured on PVG-coated microcarriers achieved six-fold expansion ability [82], and also maintained their immunopotency and differentiation capacity in xeno-free bioreactor conditions [83].

#### 3.1.2. Perfusion-Based Bioreactors

In perfusion-based bioreactors, MSCs are attached to an immobilized substrate under shear stress from perfusion, such as a packed bed [76]. A hollow fiber bioreactor is an excellent choice for scalable exosomes production. The hollow fiber bioreactor has a reservoir bottle with hollow and semi-permeable fibers, providing a large surface area for cell adhesion. Cells adhere to the surface of the hollow fibers and feed on nutrients from the medium pumped into the fibers, releasing exosomes outside the fibers [84]. It allows for continuous culture and production of exosomes over several weeks [74]. Compared with 2D culture, the exosomes of umbilical cord mesenchymal stem cells (UCMSCs) cultured in hollow fiber bioreactors (3D culture) yielded 7.5 times higher [85]. In addition, 3D-cultured MSC-Exos exhibited stronger biological functions, which were reflected in the enhancement of chondrocyte proliferation and migration, inhibition of cell apoptosis in vitro, as well as alleviation of cartilage degradation in vivo [85]. Moreover, the hollow fiber bioreactor combined with tangential flow filtration (TFF) and SEC can be useful for the scalable production of highly purified and bioactive exosomes [86].

### 3.2. Increasing Single Cell Secretion of Exosomes

In order to adapt to the environment, cells can regulate the release amounts of exosomes to convey stress information. Therefore, artificially changing the cultural environment of cells can increase the production of exosomes (Figure 2B), such as glucose starvation [87], hypoxia [88,89,90], low pH [91], heat stress [92], radiation [93], photo-activation [94], low-intensity ultrasound [95] and near ultraviolet (UV) at 365 nm [96]. These methods can change the composition and affect the function of exosomes. Exosomes from hypoxia-treated MSCs possessed significantly higher levels of microRNA (miR)-210 and enhanced cardioprotective effects [89]. In addition, hypoxia strengthened the antitumor effect of NK cell-derived exosomes [90]. Low-intensity pulsed ultrasound enhanced the promoting effect of human bone marrow mesenchymal stem cell (BMSC)-derived exosomes on cartilage regeneration via modulating the noncanonical nuclear factor-kappaB (NF-κB) signaling pathway [97].

Moreover, the yield of exosomes can be improved by chemical treatment. Alcohol [98], photosensitizer [94], doxorubicin [94], gemcitabine [99], melphalan [100], tetramethylpyrazine [101], paraformaldehyde [102], dithiothreitol [102], palmitic acid [103] and iodoacetate plus 2,4-dinitrophenol [104] have been shown to stimulate exosome release.

There are also some engineering materials and designs to trigger exosome release. Patel et al. fabricated a 3D-printed scaffold-perfusion bioreactor system that significantly enhanced exosome production yield by over 100-fold compared to tissue culture flasks [105]. The shear stress generated by the flow rate they set simulated that which is generated under physiological conditions in vivo [105]. Hisey et al. proposed an easy and inexpensive method for exosome production by designing polystyrene microtracks over a 100 mm diameter growth surface area [106]. The microtrack patterning increased not only cell growth density, but also the number of exosomes produced per cell [106]. Chen et al. used porous gelatin methacrylate (Gelma) hydrogel for producing 3D-exosomes, the yield of which was approximately 3.68 to 6.64-fold higher than 2D-exosomes [107]. Utilizing this hydrogel was demonstrated to increase yield of exosomes released per cell and reduce costs by reducing culture space and medium volume, which was promising for mass production [107].

On the other hand, we can also achieve mass production of artificial exosomes by breaking the cells and recombining the released cellular components (Figure 2B). These methods include nitrogen cavitation [108], extrusion via porous membrane [109,110], sonication [111] and high pH solution plus sonication [112].

## 4. Preparation of Engineered Exosomes

To overcome the drawbacks of natural exosomesm such as weak function and low targeting, there remains the need to engineer exosomes. The engineering preparation methods of exosomes are mainly divided into pretreatment of parental cells, drug loading and surface modification (Figure 3A). In these aspects, we pay more attention to how to improve the activity and target ability of exosomes, and how to add therapeutic drugs into exosomes. Pretreatment can enhance the innate activity of exosomes; furthermore, genetic or phenotypic modification of the parent cells and direct processing of exosomes can improve the function of the exosomes as drug carriers with a high target ability.

### 4.1. Pretreatment of Parental Cells

Pretreatment methods mainly include altering the cell growth environment and adding biochemical factors. External stimuli can modulate the physiological state of the parental cell, thereby affecting the biological function of the exosomes it produces. For instance, MSCs are hypoxic in vivo but are exposed to the air in a culture medium. The absolute pO_2_ of the bone marrow is quite low (<32 mmHg) despite high vascular density [113]. Hypoxia preconditioning enhanced the effect of exosomes on bone fracture healing as well as increased the release of exosomes [114]. Mechanically, hypoxia could strengthen the function of exosomes by microRNAome alterations in MSC-Exos, such as hsa-miR-181c-5p, hsa-miR-18a-3p, hsa-miR-376a-5p and hsa-miR-337-5p [115]. MiR-18-3p and miR-181c-5p were involved in janus kinase (JAK)/signal transducer and activator of transcription (STAT) and MAPK signaling pathway, respectively, which explained that hypoxia-preconditioned MSC-Exos promote cartilage repair to a greater extent than normoxia-preconditioned MSC-Exos [115]. Moreover, exosomes released from kartogenin-pretreated human UCMSCs were found to have the potential to induce chondrogenic differentiation with abundant miR-381-3p, but not in exosomes from the untreated cells [116]. Kartogenin enabled BMSC-derived exosomes to exhibit a more significant promoting effect on cartilage matrix formation [117]. Exosomes from MSCs treated with curcumin played a stronger role than the control group in attenuating OA progress [118]. Inflammatory cytokines such as IFN-γ and tumor necrosis factor alpha (TNF-α) stimulated ADSCs to release immunosuppressive exosomes [119]. TGF-β1, used to stimulate MSCs, led to the high expression of miR-135b in MSC-Exos and the regeneration of cartilage tissue [120,121].

### 4.2. Drug Loading

Loading drugs into exosomes can allow for enhanced therapeutic effects. Drug loading methods are mainly divided into two categories: the modification of parental cells or directly drug loading into exosomes after secretion, which are called endogenous and exogenous loading methods, respectively [122,123]. MicroRNAs and proteins with therapeutic benefits are usually loaded into exosomes by endogenous loading methods, while small molecule drugs are loaded exogenously [35].

#### 4.2.1. Endogenous Loading Methods

Target molecules can be loaded into exosomes endogenously through intracellular cargo sorting mechanisms. We can enrich therapeutic drugs in parental cells to secrete engineered exosomes, using lentiviral-based systems [124,125,126,127] and Lipofectamine 2000 [128,129,130] or Lipofectamine 3000 [131] for transfection. Pan et al. demonstrated that significantly increased levels of miR-132-3p in MSC-Exos after transfection of MSCs with lentivirus contributed to ameliorating brain ischemic injury [126]. Using the lentivirus infection, ADSCs overexpressing miR-138-5p were generated to secrete miR-138-5p-enriched exosomes, which was promising for bladder cancer treatment [127]. With Lipofectamine 2000, human UCMSCs were transfected with a miR-100-5p inhibitor and a miR-100-5p mimic to achieve miR-100-5p reduction and overexpression in their exosomes, respectively [130].

For better entry into exosomes, target proteins can be modified such as ubiquitination [132] and tagging with a WW tag [133]. The WW tag means a triple-stranded β sheet domain consisting of 38 to 40 amino acid residues, named for the characteristic that contains two tryptophanes (Trp, W) [134]. The WW tag can be recognized by the late-domain motifs on Ndfip1, leading to the package of target proteins into exosomes [133]. Another strategy is to find a guide molecule that loads the target cargo into exosomes, such as constitutive proteins or viral proteins known to be able to enter exosomes. For proteins, fusion with the exosome-anchoring protein Nef and a vesicular stomatitis virus glycoprotein (VSVG) increased its levels in exosomes [135,136]. A technique called exosomes for protein loading via optically reversible protein-protein interaction (EXPLORs) and using blue light significantly improved the loading efficiency of therapeutic proteins via optically reversible protein–protein interaction [137]. For nucleic acid, it requires intermediate to interact with the guide protein. Kojima et al. developed exosomal transfer into cells (EXOtic) devices for actively packaging specific mRNAs into exosomes, bounding an RNA binding protein L7Ae to the C-terminus of CD63, which could interact with the C/D_box_ inserted into the mRNA [138]. With the help of Lamp2a fusion protein, the modified miR-199a realized its enrichment in exosomes via TAT peptide/TAR RNA loop interaction [139]. The interaction between RNA aptamer MS2 and MS2 bacteriophage coat protein (MCP) combined with blue light-mediated reversible CIBN-CYR2 interaction were introduced to enrich the miR-21 sponge RNA into exosomes [140]. After fusion with CD9, human antigen R (HuR) could attract miR-155 into exosomes [141].

#### 4.2.2. Exogenous Loading Methods

To obtain engineered exosomes for Parkinson’s disease therapy, mechanical methods were applied to loading catalase such as the incubation at room temperature, permeabilization with saponin, freeze-thaw cycles, sonication or extrusion [142]. It should be noted that the integrity and activity of exosomes produced by mechanical methods are easily affected, so corresponding parameters need to be adjusted. To functionalize macrophage-derived exosomes, coextrusion via a liposome extruder (220 µm, 12 times) was applied to package panobinostat and p53-induced protein phosphatase 1 (PPM1D)-siRNA into exosomes for glioma treatment [143].

Co-incubation allows therapeutic drugs to enter into exosomes directly, such as microRNAs [144], plasmids [145], siRNAs [146], lipophilic drug molecules (e.g., curcumin [147,148], paxlitaxel [149,150]) and water-soluble molecules (e.g., antocyanidins [151]). Hybrid exosomes were generated by incubating the original exosomes with liposomes, with the encapsulation of large plasmids [145]. However, co-incubation is inefficient and requires large amounts of therapeutic drugs. Mild electroporation is a more efficient method for loading microRNAs and better protects microRNAs from ribonuclease (RNase) degradation than co-incubation [144].

Electroporation is an easy-to-operate method, but it leads to aggregation of exosomes, leakage of endogenous cargo and disruption of protein activity. In the electric field, temporary pores in the exosome membrane are opened by short, high-voltage pulses. Electroporation allowed a locked nucleic acid (LNA)-modified anti-miR-142-3p oligonucleotide to enter the exosomes [152]. Doxorubicin, a kind of hydrophilic small-molecule cytotoxin, was reported for loading into exosomes by an improved electroporation method, which preserved the integrity and intended function of exosomes as well as improved loading efficiency, recovery and drug potency [153]. It was showed that the potency of doxorubicin in exosomes was 190-fold higher than doxorubicin alone in vitro, and twice the potency of a liposomal form of doxorubicin [153].

Sonication is also an efficient method for loading paclitaxel into exosomes with a high drug loading capacity [154,155]. To load vancomycin, the purified vancomycin-mixed exosomes were sonicated and then incubated at 37 °C for 60 min to recover the exosome membrane [156]. The lysostaphin-exosome mixture was also sonicated to produce lysostaphin-loaded exosomes [156].

Chemical reagents such as calcium chloride and PEG can also be used for exogenous drug loading [157,158]. PEG mediates full fusion between exosomes and liposomes containing various components without leakage to generate engineered exosomes of various components, such as a lipophilic fluorescent probe, a hydrophilic Rhodamine probe and a fluorescent clinically approved antitumor photosensitizer mTHPC [158]. This system holds promise for loading any compound associated with synthetic liposomes into exosomes, whether they have hydrophilic compounds inside (e.g., RNA, nanoparticles, drugs and imaging contrast agents) or lipophilic compounds on the surface (e.g., membrane proteins, targeting agents, PEGylated lipids) [158].

### 4.3. Surface Modification

The target ability is important for exosomes to exert their therapeutic function. Common strategies can be designed from both endogenous modification in parent cells and direct modification of exosome surface molecules.

Targeting ligands are added onto the surface of exosomes to achieve precise delivery of therapeutic drugs to lesions. Derived from lentivirus-transfected cells, exosomes with Lamp2b linked to a cardiomyocyte specific peptide at the N-terminus were able to enhance cardiac tropism and reduce cardiomyocyte apoptosis. To express rabies virus glycoprotein (RVG) on the exosomes, pcDNA3.1(−)-RVG-Lamp2b plasmids were transfected into MSCs, and the produced exosomes could deliver miR-124 to the infarct site [159]. Klotho-modified exosomes derived from plasmid-transfected MSCs could effectively target circulating endothelial progenitor cells (EPCs) [160]. The metabolic engineering and labeling approach combined with bio-orthogonal click chemistry can conveniently modify and functionalize exosomes without complicated genetic fusion processes [161]. L-Azidohomoalanine (AHA) was co-cultured with cells and involved in protein metabolism to isolate azide-integrated exosomes [156]. Strain-promoted azide-alkyne click chemistry reaction (SPAAC) allowed dibenzocyclooctyne (DBCO)-mannosyl ligands to react with azide to obtain mannose-modified exosomes, with no effect on the structure and function of exosomes [156]. Mannosylated exosomes would target to macrophages expressing high levels of mannose receptors and subsequently transport to the site of intracellular methicillin-resistant Staphylococcus aureus (MRSA) infection to deliver antibiotics [156].

In addition to the target ability derived from secretory cells, direct surface modification is a promising option to artificially enhance targeting. CP05 could anchor various targeting peptides on exosomes by binding to CD63, helping exosomes selectively deliver to target cells [162]. Using bio-orthogonal copper-free click chemistry to conjugate the cyclo(Arg-Gly-Asp-D-Tyr-Lys) (c(RGDyK)) peptide onto exosome surfaces allowed the engineered exosomes to target the lesion region of the ischemic brain [163]. By post-insertion technology, modification of c(RGDyk) peptide in paclitaxel-loaded exosomes enhanced targeting and significantly improved the efficacy in glioma therapy [149]. The aminoethylanisamide (AA)-PEG-vectorized exosomes loaded with paclitaxel could target the sigma receptor and improve anti-lung cancer outcomes [155]. Exosomes combined with various nanoparticles is a novel targeted system for drug delivery. A33-positive exosomes loaded with doxorubicin could bind to high-density A33 antibodies on nanoparticles and form an antitumor complex targeting A33-positive colon cancer cells [164]. RNA nanoparticles were anchored on the exosomes membrane with cholesterol that was attached to the arrowtail of pRNA-3WJ, providing three classes of surface targeting ligands for prostate-specific membrane antigen (PSMA), epidermal growth-factor receptor (EGFR) and folate receptor [165].

## 5. Therapeutic Strategies of Engineered Exosomes in OA Treatment

MSC exosomes derived from different tissues have been demonstrated to be beneficial for OA treatment, with diverse active components and distinct mechanisms. Therapeutic exosomes can be obtained from embryonic stem cell-induced mesenchymal stem cells (ESC-MSCs) and MSCs derived from bone marrow, umbilical cord, synovium, adipose, etc. Different tissue origins also affect the efficacy of exosomes [166]. It is worth noting that other sources of exosomes play an important role in OA progression or treatment, including but not limited to chondrogenic progenitor cells [167], OA subchondral bone [168], platelet-rich plasma [169] and vascular endothelial cells [170].

### 5.1. Active Components within Exosomes for OA

ESC-MSCs has been shown to be an optional source of therapeutic exosomes. Similar to ESC-MSCs, intra-articular injection of ESC-MSC-derived exosomes successfully protected cartilage from damage in the destabilized medial meniscus (DMM)-induced OA mice model [171]. Under ESC-MSC-derived exosomes treatment in vitro, the chondrocyte phenotype was maintained with an increased level of collagen type II and a decrease in ADAMTS5 expression in exposure to interleukin 1 beta (IL-1β) [171]. Furthermore, exosomes from ESC-MSCs promoted chondrocyte proliferation and migration, which was related to the exosomal CD73-mediated activation of pro-survival protein kinase B (AKT) and extracellular signal-regulated kinase (ERK) signaling pathways via adenosine [172].

The umbilical cord is a clinical feasible origin for MSCs and their exosomes, with a painless collection process. Exosomes from human UCMSCs demonstrated the facilitation effect on proliferation, migration and differentiation of chondrocytes and BMSCs, via activation of the phosphatase and tensin homologue (PTEN)/AKT signaling pathway by miR-23a-3p [30]. Moreover, human umbilical cord Wharton’s jelly mesenchymal stem cell (hWJMSC)-derived exosomes could modulate inflammation in the joint cavity by inducing polarization of macrophages toward the M2 phenotype [18].

Human synovial mesenchymal stem cell (SMSC)-derived exosomes attenuated IL-1β-induced apoptosis, degeneration and degradation in chondrocytes as well as inflammation process [29,173]. Exosomal miR-129-5p and miR-212-5p targeted high mobility group protein-1 (HMGB1) and E74-like factor 3 (ELF3), respectively, partially accounting for the mechanisms [29,173]. Another potential functional microRNA is miR-155-5p targeting runt-related transcription factor 2 (Runx2) [174]. SMSC-derived exosomes promoted proliferation and migration, inhibited apoptosis of chondrocytes, but could not promote extracellular matrix (ECM) secretion until overexpression of miR-155-5p, effectively preventing OA in mouse [174].

Bone marrow mesenchymal stem cell derived exosomes (BMSC-Exos) have been found to exert therapeutic effects on OA, both in vivo and in vitro. In vitro, it was verified that IL-1β-induced senescence and apoptosis of chondrocytes can be attenuated by BMSC-Exos [15]. After internalization of BMSC-Exos, the migration and secretion of chondrocytes could be enhanced with the upregulation of collagen type II, aggrecan and SRY-box transcription factor 9 (SOX9) expression and downregulation of matrix metallopeptidase 13 (MMP-13) expression though exosomal miR-136-5p inhibiting ELF3 [26]. In vivo, BMSC-Exos not only protected cartilage from degradation [26,175], but also achieved subchondral bone remodeling, involving the trabecular bone volume fraction, trabecular number and connectivity density [15]. After exosome treatment, knee pain was effectively relieved in OA rats [176]. The running capacity and cartilage tissue damage of the OA mice were also improved, with increased chondrocyte glutamine metabolism by regulating c-MYC [27]. Furthermore, exosomes from BMSCs exerted an inhibitory effect on inflammation to alleviate OA by regulating inflammatory factors such as TNF-α and interleukin-6 (IL-6) [27,28], reducing oxidative stress injury [28] and stimulating macrophage polarization towards anti-inflammatory M2 phenotype [175]. The mechanism was partially associated with exosomal miR-9-5p targeting syndecan-1 [28].

Qi et al. conducted a study on the tissue origin effect of exosomes, comparing three types of exosomes from BMSCs, ADSCs and SMSCs [166]. It was demonstrated that ADSC-derived exosomes exhibited the highest efficiency in promoting proliferation, migration and chondrogenic differentiation of BMSCs in vitro as well as cartilage tissue regeneration in vivo [166]. Moreover, ADSC-derived exosomes showed the potency to decrease local inflammation and induce chondrogenesis of periosteal cells in vitro, the mechanism of which was relevant to increased levels of miR-145 and miR-221 [31]. ADSC-derived exosomes could also affect gene expression and protein release of both chondrocytes and synoviocytes, weakening an IL-1β-induced inflammatory response [177]. Quantitative proteomics analysis revealed the differences in protein profiles, with a set of higher protein content in ADSC-derived exosomes associated with focal adhesion, ECM-receptor interaction, phosphatidylinositol-3 kinase (PI3K)/AKT signaling pathway, etc. [166].

Adipose can serve as an easily obtained, patient-specific and theoretically abundant tissue source of MSC-Exos in the treatment of OA [31]. Clinically, infrapatellar fat pad (IPFP) is more convenient and feasible to obtain from OA patients during arthroscopic operation [178]. As well as suppression of chondrocyte apoptosis and promotion of ECM synthesis in vitro, MSC-Exos from IPFP was proven to improve gait abnormality in OA mice though maintaining cartilage homeostasis, which was due to enhanced chondrocyte autophagy via miR-100-5p-mediated inhibitory mechanistic target of rapamycin (mTOR) signal [178].

### 5.2. Exosome-Based Drug Loading Strategies for OA

The efficacy of exosome-based treatment depends largely on the cargo it carries. An increasing amount of research has revealed that non-coding RNAs, proteins and small molecule drugs can serve as crucial regulators in cartilage metabolism, and also promises therapeutic targets for the treatment of OA [179,180,181,182,183,184]. With the capability of carrying lipid, protein and nucleic acid, exosomes can serve as an optimal vehicle for OA drug delivery (Figure 3A) [157].

#### 5.2.1. MicroRNAs

Natural exosomes derived from human SMSCs promoted the proliferation and migration of articular chondrocytes while decreased ECM secretion via yes-associated protein (YAP) activated by Wnt5a and Wnt5b [124]. Engineered exosomes can artificially avoid side effects of natural exosomes and enhance their therapeutic efficacy. Using lentiviral-based systems, miR-140-5p was overexpressed in SMSCs and its exosomes, which could retain positive effect in chondrocytes, alleviate the repressive effect of YAP on SOX9 and ECM secretion though RalA, and prevent OA in rat model [124]. Wnt is known as a key molecule involved in the pathogenesis of OA. Mao et al. demonstrated that exosomes derived from miR-92a-3p-overexpressing human MSCs promoted chondrogenesis, maintained the function of articular chondrocytes and inhibited cartilage degradation in the OA mice model [185]. A series of experiments indicated that exosomal miR-92a-3p directly targeted the 3′-UTR of Wnt5a mRNA and suppressed its expression, exhibiting potential as a Wnt inhibitor for OA treatment [185].

Delivering microRNAs by exosomes is of significance for the treatment of OA. MSC-Exos-derived miR-135b promoted chondrocyte proliferation and cartilage repair by regulating Sp1 [121], as well as improved cartilage damage in OA rats through the enhancement of M2 synovial macrophages polarization via inhibiting MAPK6 [120]. Overexpressed miR-26a-5p in BMSC-Exos was transferred into synovial fibroblasts to suppress cell proliferation and migration, regional inflammation, and induce cell apoptosis by downregulating PTGS2, thus exerted an alleviatory effect on OA damage in vitro and in vivo [125]. BMSC-Exos could deliver miR-326 to chondrocytes and inhibited pyroptosis of chondrocytes by targeting HDAC3 and activating the STAT1/NF-κB p65 signaling pathway [186]. As demonstrated by in vivo experiments, overexpressed miR-326 ameliorated the pathogenesis of OA [186]. BMSC-derived exosomal miR-125a-5p was found to upregulate the expression of collagen type II, aggrecan and SOX9 but MMP-13 via targeting E2F2, accelerating chondrocytes migration in vitro and inhibiting cartilage degeneration in post-traumatic OA mice model [187].

MicroRNAs have been consistently proven to regulate chondrocyte proliferation, differentiation, migration, apoptosis and matrix synthesis or degradation, such as miR-7 [188], miR-195 [189], miR-103 [190], miR-455-3p [191] and miR-33b-3p [192]. These microRNAs play different physiological functions and mediate different signaling pathways. Whether there will be better effects through the combined application of microRNAs and whether the side effects can be reduced through the adjustment of microRNAs loading deserve further research and exploration.

#### 5.2.2. Long Non-Coding RNAs

Long non-coding RNAs (lncRNAs) in the exosomes play an important role in the therapeutic functions for OA. Exosomal lncRNA MEG-3 could maintain the chondrocyte phenotype by upregulating the expression of collagen type II and inhibiting IL-1β–induced senescence and apoptosis in chondrocytes [15]. Liu et al. established that MSC-Exos upregulated collagen type II alpha 1 chain (Col2a1) and aggrecan, downregulated MMP-13 and Runx2, and significantly reversed IL-1β-induced chondrocyte proliferation inhibition and apoptosis induction, the crucial effective molecule in which was demonstrated to be lncRNA KLF3-AS1 [193]. Furthermore, lncRNA KLF3-AS1 acted as a miR-206 sponge to motivate the expression of GIT1 in chondrocytes, reducing chondrocyte injury [194]. In addition, LncRNA MCM3AP-AS1 [195], DNM3OS [196], SNHG1 [197], DANCR [198] and SNHG5 [199] were also demonstrated as potential therapeutic targets for OA, and as promising exosome-loading drugs.

#### 5.2.3. Circular RNAs

Accumulating evidence has indicated the key role of circular RNAs (circRNAs) in OA. Mao et al. demonstrated that upregulation of exosomal circRNA_0001236 enhanced the expression of Col2a1 and Sox9 via sponging miR-3677-3p to balance ECM anabolism and catabolism, thus suppressing OA progression in the DMM mice [200]. Although the sample size was small, this study provided clear evidence that a novel exosomal circRNA_0001236 could play an important role in chondrogenic differentiation and a potentially effective therapeutic strategy for treating OA. CircHIPK3 could sponge miR-124-3p directly and upregulate the expression of MYH9, accounting for MSC-Exos-mediated chondrocyte proliferation and migration induction and chondrocyte apoptosis inhibition [201]. Consequently, Exosomal circHIPK3 distinctly relieved IL-1β-induced chondrocyte injury [201].

As for the metabolism of ECM, circSERPINE2 promoted synthesis and suppressed degradation [202], while circRNA.33186 did the opposite [203]. Overexpression of circSERPINE2 and knockdown of circRNA.33186 were proven to alleviate OA [202,203]. The effect of loading them into exosomes remains to be further explored.

#### 5.2.4. Proteins

Wnt is an important protein family with a high degree of conservation and evolutionary constraint, leading to the activation of β-catenin/TCF target genes [204]. Wnt3a, a molecule upregulated by acute cartilage injury, was conducive to cartilage regeneration; however, the direct injection could not achieve efficient penetration of recombinant Wnt3a through cartilage ECM into chondrocytes [180]. It was discovered that exosomes carrying Wnt3a effectively activated Wnt signaling in cartilage for more than one week and promoted the healing of osteochondral defects for a long time in mice [180].

TGF-β superfamily and bone morphogenetic proteins (BMPs) were considered as important regulators for cartilage formation [205,206]. TGF-β3- and BMP-6-induced multipotent stem cells exerted increased chondrogenic effect in an OA sheep model [181]. MSC-Exos are being developed as drug delivery vehicles with a combination of TGF-β3 and BMP-6 for OA treatment [207].

#### 5.2.5. Small Molecule Drugs

A small molecule kartogenin was able to induce chondrogenic differentiation of SMSCs but tended to precipitate in cells due to low water solubility [208]. To maintain its effective concentration in cells, kartogenin was loaded into exosomes with MSC-binding peptide E7 (E7-Exos), which displayed excellent performance on cartilage differentiation in vitro and in vivo [208].

Curcumin is a kind of natural polyphenol compound extracted from rhizomes of the plant *Curcuma longa* and has anti-inflammatory effects in knee OA rats by blocking the TLR4/NF-κB signal pathway [182]. A double-blind multicenter randomized placebo-controlled trial indicated that bio-optimized Curcuma longa extract was an efficient and safe treatment to relieve pain in knee OA patients [183]. Curcumin was encapsulated in BMSC-Exos after incubating BMSCs with curcumin [184]. Compared with control BMSC-Exos and free curcumin, curcumin-containing exosomes significantly enhanced the viability and migration ability of osteoarthritic chondrocytes and inhibited their apoptosis [184]. The elevated expression of miR-126-3p induced by exosomal curcumin suppressed phosphorylation of Erk1/2, PI3K/Akt and p38 MAPK, thus negatively regulating pro-inflammatory signaling pathways [184].

### 5.3. Biological Materials for Exosome Delivery

The most common method of exosome delivery in OA treatment is intra-articular injection [14]. Due to the swift clearance of exosomes, it is necessary to make them more functionally stable during the long healing process of cartilage. While multiple intra-articular injections of exosomes increase pain and the risk of infection, the frequency of exosome injection together with hyaluronic acid can be reduced to three times a week, which is more clinically acceptable [25,209]. To achieve targeted localization of exosomes at cartilage defect sites and maintain their durable activity, biomaterials such as hydrogels and ECM-derived scaffolds have been utilized (Figure 3B).

#### 5.3.1. Hydrogel

In order to better apply exosomes in vivo, it is important to select a material with excellent biocompatibility, mechanical property and biodegradability for the sustained release of exosomes. Hydrogel is a promising biomaterial in cartilage tissue engineering with its injectable and UV-cross-linked properties. Gelma/nanoclay hydrogel (Gel-nano)- containing exosomes effectively stimulated chondrogenesis and promoted cartilage regeneration [30]. As the hydrogel degraded, exosomes were released continuously and then gradually internalized by chondrocytes and BMSCs. The internalized exosomes promoted cellular migration, proliferation and chondrogenic differentiation, leading to the synthesis of glycosaminoglycan, collagen type II and ECM [30]. Tao et al. applied poly(D,l-lactide)-b-poly(ethylene glycol)-b-poly(D,l-lactide) (PDLLA-PEG-PDLLA, PLEL) triblock copolymer gels to serve as injectable carriers of circRNA3503-loaded exosomes [210]. Through sponging hsa-miR-181c-3p and hsa-let-7b-3p, circRNA3503-loaded exosomes had a significant promoting effect on chondrocyte renewal and ECM production to prevent OA progression [210]. Liu et al. invented a promising in situ tissue patch to achieve highly effective cartilage repair and regeneration by one time implantation, integrating MSC-Exos with a photoinduced imine crosslinking (PIC) hydrogel glue [211].

#### 5.3.2. ECM-Derived Scaffolds

Exosomes can synergize with bioactive scaffolds to exhibit enhanced tissue repair function. Recently, ECM-derived scaffolds have gained extensive attention in tissue engineering due to their low immunogenicity, high biocompatibility and good biodegradability [18,212,213]. The acellular cartilage extracellular matrix (ACECM) scaffold minimizes immune rejection in vivo due to removal of cells by decellularized technology. It had been demonstrated to be an effective bioactive material scaffold for hyaline cartilage repair and subchondral bone reconstruction [213]. In addition, the reparative effect of the ACECM scaffold was enhanced by hWJMSC-derived exosomes in a rabbit osteochondral defect model [18]. Mechanically, the ACECM scaffold provided a cartilage-like microenvironment facilitating local cell attachment, proliferation and chondrogenesis [213]. Meanwhile, hWJMSC-derived exosomes could stimulate osteochondral regeneration as well as regulate the microenvironment of the articular cavity by inhibiting the inflammatory response and promoting cartilage ECM synthesis via exosomal microRNAs such as miR-148a and miR-29b [18]. With the help of desktop-stereolithography technology, a 3D printed ECM/Gelma/exosome scaffold with radially oriented channels went from concept to reality, providing a more convenient one-step operating system to effectively control the delivery of exosomes [214]. This 3D-printed ECM/Gelma/exosome scaffold effectively expedited cartilage regeneration in a rabbit model through inducing mitochondrial biogenesis, chondrocyte migration and M2 macrophage polarization [214].

## 6. Discussion

For early-stage OA patients, non-surgical therapies are considered first, with the goal of relieving pain or improving physical conditions without restoring cartilage, and then surgery is recommended if non-surgical treatments fail to manage symptoms [4]. Nonpharmacological therapies including exercise, weight loss and education. Non-steroidal anti-inflammatory drugs (NSAIDs) are the initial oral medication of choice in OA treatment [215]. However, adverse effects of OA medications have restricted their application, including cardiovascular, gastrointestinal and renal problems, which are common comorbidities in OA population [5]. An intra-articular injection of corticosteroid has been demonstrated to exert positive effects on relieving pain over placebo [5], with adverse events such as accelerated OA progression and joint destruction [216]. Recently, Wnt pathway inhibitor (e.g., SM04690) [217], recombinant human fibroblast growth factor 18 (e.g., Sprifermin) [218] and cathepsin K inhibitor (e.g., MIV-711) [219] are emerging as disease-modifying OA drugs (DMOADs) undergoing clinical trials.

For end-stage OA patients with persistent pain, functional loss and advanced radiographic changes, joint replacement surgery is an effective treatment option [3,220], but it also has disadvantages such as infection risk, short service life of materials and the possible need for secondary surgery [6]. To repair damaged cartilage, there are traditional surgeries including subchondral bone microfracture and soft tissue grafts [221], and many advanced techniques such as particulated juvenile allograft cartilage (PJAC) [222] and autologous matrix-induced chondrogenesis (AMIC) [223]. The common challenges for cartilage tissue engineering that often arise after transplantation into the body are the production of fibrous cartilage instead of hyaline cartilage, poor integration and inflammation [221].

Regenerative medicine based on MSC therapy has opened new avenues for OA treatment with high expectations [11]. Some main concerns include genetic instability and chromosomal aberration caused by in vitro cultures and long-term chemotherapy or radiotherapy, which increase the risk of tumorigenesis [16]. The potential risk and the underlying mechanisms of MSC therapy require further detailed exploration. Research begins to move towards MSC-Exos as a cell-free alternative to MSC therapy, which can recapitulate most of the therapeutic functions of parental MSCs [224]. MSC-Exos have been demonstrated to effectively accelerate cartilage repair though regulating a series of biological processes, such as promoting chondrocyte proliferation, migration and matrix synthesis, as well as reducing apoptosis and immune reactivity, thus exhibiting great therapeutic potential for OA [172].

Different sourced (e.g., immune cell, blood cell, neural cell) exosomes possess their unique biological functions and tendency to cure certain diseases, on the basis of their specific properties and cargoes [225]. Compared to other sources, MSC is a popular choice because of the convenience of isolation and specialized biological functions [32]. As pluripotent and prolific producers of exosomes, MSCs can produce exosomes containing bioactive components related to self-renewal and differentiation [225]. However, the best source of MSC-Exos remains unclear. It is also essential to continue exploring the clinical potential of other sourced exosomes.

There have been several reviews describing the pathogenic roles and therapeutic potential of exosomes in OA and summarizing related engineering strategies, that have provided solid evidence that exosomes can serve as a promising cell-free therapy for OA [226,227,228]. Given the hurdles encountered in the clinical translation of exosomes, we focus more on detailed techniques for large-scale exosome production and preparation of engineered exosomes, and summarize treatment ideas including not only drug loading in exosomes, but also the assistance of biological materials. The purpose is to review the current technical methods and therapeutic strategies, and strive to promote the realization of the clinical application of exosomes in OA.

Clinical translation of exosome therapy requires large-scale production while ensuring functional integrity to meet therapeutic needs. Fortunately, there have been several physical, chemical approaches and bioreactors that effectively improve exosome production. Bioreactors combined with 3D-printed scaffolds enabled simulation of 3D dynamic physiological environments for the growth of cells and the secretion of exosomes, whose yield was remarkably elevated under the shear stress [105]. The mechanism was partially attributed to YAP-mediated mechanosensitivity [229]. There also exists the laborious and costly problem if the exosomes need to be modified on a large scale through genetic engineering. With the help of external devices such as an atmosphere control device, rotary system and ultrasonic equipment, it becomes feasible to subject the cells in the bioreactor to hypoxia [90,115], mechanical stimulation [230] and low-intensity pulsed ultrasound [97], which can increase exosome production while regulating the function of released exosomes.

To overcome the limitation of natural exosomes, we can engineer exosomes to enhance their biological function through pretreatments of parental cells, drug loading and surface modification methods. Since natural exosomes are produced through paracrine, their composition is similar to that of parent cells and varies with their physiological changes. Therefore, the cargoes and function of exosomes can be easily controlled by pretreatments such as physicochemical stimuli. However, considering that MSCs are in a dynamic state of stress, the optimal pretreatment parameters and combinations of various factors need to be further explored. Furthermore, it remains unclear whether exosomes derived from cellular stress have side effects other than the therapeutic effects [231].

Another future challenge for engineering exosomes is improving drug-loading efficiency. The limited loading space is the major difficulty for exogenous drugs to enter the exosomes that originally contain cargoes from their parent cells, which might be the reason that the drug loading efficiency of exosomes is lower than that of liposomes [32]. Usually, endogenous methods are used to package microRNAs and proteins into exosomes and exogenous approaches are for therapeutic drugs and small molecules. However, the loading efficiency of different methods varies significantly. In a previous study, the encapsulation rates of the direct incubation of kartogenin with exosomes, repeated freeze-thaw procedure and electroporation were 8%, 13% and 40%, respectively [208]. Therefore, it is of importance to optimize existing techniques and explore new methods to achieve higher drug loading efficiency. A new direction is to take advantage of nanoparticles [232,233].

As for the target ability of exosomes, surface modification can enhance targeting of exosomes to specific tissues, and biomaterial-assisted delivery can also improve anatomical targeting. Given the complexity and the low transfection rate of genetic modification of parent cells, bio-orthogonal copper-free click chemistry offers an easy-to-operate method with higher efficiency and reproducibility while avoiding the toxicity of copper(I) to cells [234]. The following limitations also need to be considered: surface structure alteration of exosomes, impact on biological activity of membrane proteins and the safety of modification by virus-derived proteins [53].

Future exosome engineering technology should be developed towards producing therapeutic exosomes with higher yield, biological activity, drug loading efficiency and targeting ability. In addition, the enrichment of active components that alleviate OA combined with the reduction of biofactors that contribute to OA pathogenesis might achieve more significant treatment efficacy. Some emerging technologies deserve more attention, such as bottom-up assembly of extracellular vesicles with precisely controlled lipid, protein and RNA composition, which have exhibited similar function to natural exosomes [235]. Efforts to obtain mass-produced engineered exosomes quickly and stably with customized and enhanced functions will accelerate exosomes to become part of the next-generation treatments in regenerative medicine for OA.

To date, no clinical trials have been completed to evaluate exosome-based therapy for OA. There are a few clinical trials in progress (Table 1). Accumulating preclinical trials have shown that engineered exosomes have strengthened function and targeting, confirming their therapeutic potential for OA, but lack of clinical trial validation. Insufficient clinical evidence has seriously hindered the translation of exosome-based therapy from labs to clinical application. Moreover, despite significant progress in understanding the biology of exosomes, the functional mechanism and the safety of MSC-Exos need further study. Research on exosome-based therapy for OA is still at an early stage and more clinical trials can be expected in the future.

Future research should also pay attention to the formulation of standards on production, storage and transportation of exosomes, and to find appropriate doses and the optimal time windows for administration. With the in-depth exploration of the mechanism of exosomes in OA treatment, the required cargoes can be assembled into engineered exosomes according to the individual joint conditions of patients, to achieve personalized treatment and precision medicine.

In conclusion, despite the challenges, MSC-derived engineered exosome-based cell-free therapy holds great promise in the treatment of OA and deserves further research.

## Figures and Tables

**Figure 1 membranes-12-00739-f001:**
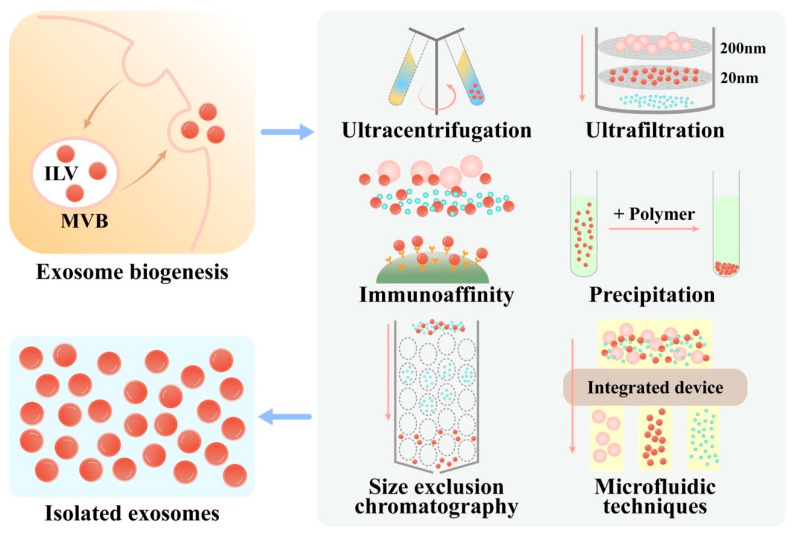
Methods for exosome isolating. ILV, intraluminal vesicles; MVB, multivesicular bodies.

**Figure 2 membranes-12-00739-f002:**
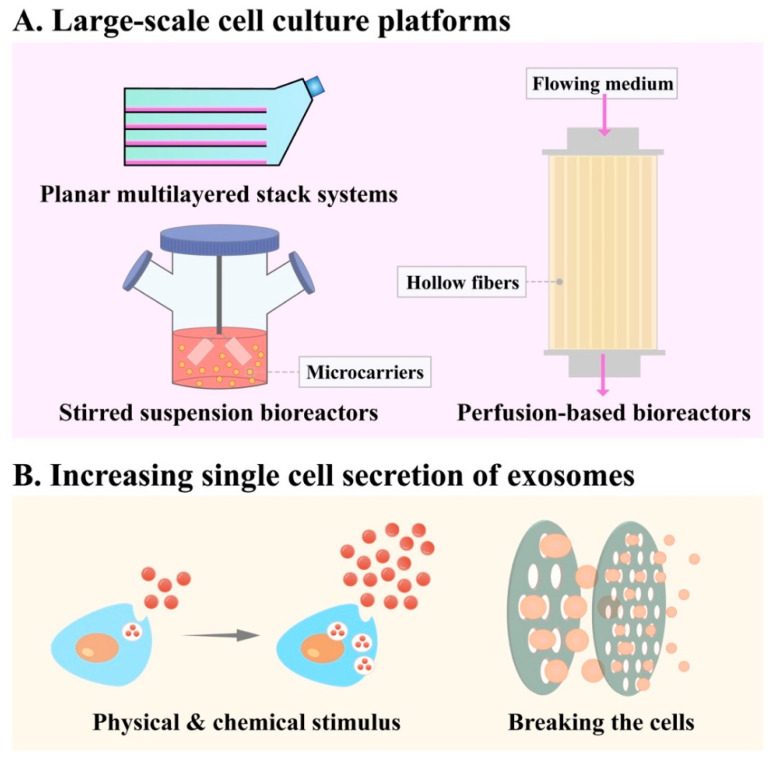
Techniques for large-scale exosome production. (**A**) Large-scale cell culture platforms. They include planar multilayered stack systems, stirred suspension bioreactors and perfusion-based bioreactors (e.g., hollow fiber bioreactor). (**B**) Increasing single cell secretion of exosomes. The methods mainly include stimulating the cells (e.g., hypoxia) and disrupting cell membranes (e.g., extrusion).

**Figure 3 membranes-12-00739-f003:**
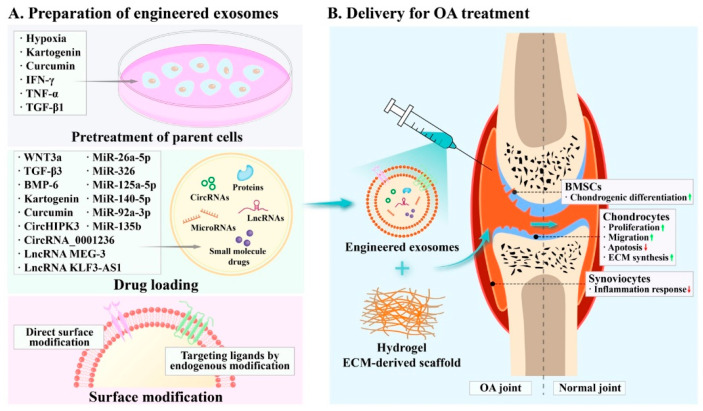
Processing and application of engineered exosomes for osteoarthritis (OA) treatment. (**A**) Preparation techniques for engineering exosomes. The processing methods include pretreatment of parent cells, drug loading and surface modification. (**B**) Application of engineered exosomes for OA. The engineered exosomes are administrated to the OA joint through intra-articular injection or scaffold implantation and exert their regulatory effects on bone marrow mesenchymal stem cells (BMSCs), chondrocytes and synoviocytes. As a result, the exosome-based therapy can suppress OA progress and accelerate cartilage regeneration to restore OA joint.

**Table 1 membranes-12-00739-t001:** Clinical trials related to secretome therapies for OA.

Study	Official Title	No. of Patients	Interventions	Study Design	Estimated Time	Phase	NCT Number
1	A Phase I Study Aiming to Assess Safety and Efficacy of a Single Intra-articular Injection of MSC-derived Exosomes (CelliStem^®^OA-sEV) in Patients With Moderate Knee Osteoarthritis (ExoOA-1)	10	Exosomes	Interventional, Single Group Assignment, None (Open Label)	October 2021–October 2023	Phase 1	NCT05060107
2	Umbilical Cord Derived Wharton’s Jelly for Treatment of Knee Osteoarthritis	12	Umbilical Cord-derived Wharton’s Jelly ^1^	Interventional, Single Group Assignment, None (Open Label)	January 2022–December 2023	Early Phase 1	NCT04719793
3	Comparative Effectiveness of Arthroscopy and Non-Arthroscopy Using Mesenchymal Stem Cell Therapy (MSCs) and Conditioned Medium From Mesenchymal Stem Cell Culture (MSCs) for Osteoartrithis With Controlled Randomization in Phase I/II	15	Mesenchymal Stem Cells with Arthoscopy|Mesenchymal Stem Cells without Arthoscopy|Conditioned Medium without Arthoscopy	Interventional, Randomized, Parallel Assignment, None (Open Label)	August 2020–December 2024	Phase 1|Phase 2	NCT04314661
4	Secretome From Mesenchymal Stem/Stromal Cells on Human Osteochondral Explants: Cocktail of Factors Secreted by Adipose-derived Stromal Cells (ASC) for the Treatment of Osteoarthritis and/or for Articular Regeneration	24	ASC secretome ^2^	Observational, Cross-Sectional	April 2021–December 2022		NCT04223622

^1^ Growth factors, cytokines, hyaluronic acid and extracellular vesicles including exosomes are all present in large quantities. ^2^ Either complete conditioned medium or extracellular vesicles.

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
