# Peer review of "Engineering of MSC-Derived Exosomes: A Promising Cell-Free Therapy for Osteoarthritis"

_membranes, 2022, doi:10.3390/membranes12080739_

Round 1
Reviewer 1 Report
In this review, Cheng et al discuss in depth various interdependent lines of research revolving around the use of exosomes in the treatment of osteoarthritis, including exosome biogenesis, detection, separation, characterization, production, modification, loading, and therapy.
The review is well written, thorough and comprehensive – an effortless and enjoyable read. I have a few suggestions the authors might consider:
- The title and introduction focus solely on exosomes, but EV-related OA research extends to other components of the secretome (see e.g. https://pubmed.ncbi.nlm.nih.gov/31689923/). Authors should probably briefly outline the broader picture, then explain their specific focus on exosomes.
- Section 2.3.1 states that UC is ‘suitable for scale-up separation’, but that’s almost the opposite of general opinion in my experience. The authors should perhaps better explain what they mean or revise the sentence.
- Section 2.3.2 states that SEC ‘fails to separate exosomes from lipoproteins’. That’s true, but all single-step techniques based on density or size fail in this respect, including UC. Authors might add this consideration to sections 2.3.1 and 2.3.2, in the context of e.g. https://pubmed.ncbi.nlm.nih.gov/28963190/
- In the same section, the authors state that “Compared to UC, SEC combined with UF generated up to 58 times more exosomes”, but I think that this result is highly linked to the specifics of reference 55 and does not have a general validity.
- In section 2.3.5, it might be worthwhile to add a mention to DLD (deterministic lateral displacement), see e.g https://pubs.rsc.org/en/content/articlelanding/2018/lc/c8lc01017j
- Section 2.4.1 might mention the fact that NTA cannot usually detect particles below 50nm in diameter. Other techniques worth mentioning in this context are dynamic light scattering and atomic force microscopy, see e.g. https://pubs.acs.org/doi/full/10.1021/acs.analchem.9b05716ù
- In section 5 it might be worth to mention other OA-related studies involving EVs, such as e.g. https://www.nature.com/articles/s41598-020-80032-7
- I’m quite sure it’s a problem linked to pdf conversion, but the small lettering in figure 1 was barely readable in the form I’ve received it.
Reviewer 2 Report
his is an interesting manuscript in which the authors review the utility, as well as limitations , of exosomes for osteoarthritis treatment. In summary, it contributes to update the state of the art on this topic. However, the authors have not included citations to relevant, recent papers (both reviews and primary research articles) that have been published on this issue. E.g. reviews PMID 31753036 PMID 32398951, or PMID 10.1097/JCMA.0000000000000570, as well as other primary articles. Please include these cites and discuss the novelty of the present manuscript compared to these recent reviews.
Reviewer 3 Report
The article providing insights into exosome-based cell-free therapy for the treatment of OA.
What are specific different approached that enhance the engineered exosomes for Osteoarthritis.
Please add one table to summarize the clinical trails related to the manuscript and how exosomes engineering is good over the naive??
If possible also add one more figure and abstract figure.
Please make it clear that what is the difference between MSC derived exosomes and benefits over the other sourced exosomes.
This paper briefly describes the current literature. I suggest to added more analysis and future directions. How it advance the current field.
Round 2
Reviewer 3 Report
No more comments